# LG-TOM: Language Grounded Theory of Mind Modeling for Multi-agent Communication

## Abstract

Emergent communication refers to the process by which multiple agents learn to develop efficient protocols for sharing information in collaborative tasks. Agents typically learn through interaction with the environment, using reinforcement learning to optimize protocols for task completion. However, the sparse task rewards can lead to unstable training and poor generalization, especially in partially observable environments and decentralized training setups. To address these challenges, we propose **LG-TOM**, a novel online RL framework that enables agents to learn from social interactions via **Theory of Mind (ToM) modeling with language grounding**. Specifically, we design a belief estimation network that leverages priors from large language models (LLMs), allowing agents to reason about how their communication influences the belief states of others. We then compute social influence as an intrinsic motivation reward, encouraging agents to share information that positively impacts teammates. Experimental results demonstrate that LG-TOM improves communication effectiveness over baselines in multi-agent collaborative tasks including complex social dilemmas.

## 1 Introduction

Multi-agent reinforcement learning (MARL) has proven effective in building systems that can solve complex problems, such as poker (Brown & Sandholm, 2019), Go (Silver et al., 2018), and real-time video games (Berner et al., 2019). A critical challenge in developing these systems is designing appropriate reward signals, where extensive prior knowledge is required to augment the relatively sparse task-completion reward with dense intermediate rewards (Ng et al., 1999). This challenge is even more critical in partially observable environments or decentralized training settings, where agent policies can struggle to converge without strong learning signals (Zhang et al., 2021).

Inspired by human evolution, where learning occurs not only from external outcomes but also from internal social dynamics (Boyd & Richerson, 2011), we explore social learning as a solution. Social learning offers a source of dense, decentralized learning signals as agents observe and reason about one another. Prior research has explored using intrinsic motivations as auxiliary rewards alongside the main task reward to encourage specific exploratory or interactive behaviors that are beneficial for collaboration. For example, curiosity-driven exploration (Pathak et al., 2017), empowerment through information maximization (Mohamed & Jimenez Rezende, 2015), and social influence rewards that encourage agents to affect each other's behavior (Jaques et al., 2019).

Theory of Mind (ToM) is the ability to infer others' hidden mental states, such as beliefs, desires, and intentions, from their observable behavior. This capacity is a cornerstone of human social interaction, collaboration, and communication (Zhang et al., 2012). Researchers have applied similar modeling techniques to improve coordination in multi-agent systems by having agents estimate the beliefs or intents of others (Wang et al., 2021; Yuan et al., 2022; Oguntola et al., 2023). However, most of this work relies on either domain-specific representations of mental states or requires direct access to other agents' internal states during training, making the methods less generalizable to novel environments and decentralized settings. The recent rise of Large Language Models (LLMs) offers a novel solution to these problems. LLMs provide a powerful, task-agnostic representation space (e.g., word embeddings) for belief inference and can approximate mental states using their zero-shot social reasoning capabilities (Li et al., 2023). In this work, we build on these ideas by

equipping agents with a language-grounded Theory of Mind (LG-ToM) to construct their social learning rewards.

Communication is critical for collaboration. Effective communication relies on a common understanding between speaker and listener about the meaning of messages. There is a line of work on emergent communication in which artificial agents develop their own communication protocols from scratch by optimizing task-specific rewards (Lazaridou et al., 2016; Lazaridou & Baroni, 2020; Karten et al., 2023a;b). However, optimizing only for task utility can lead to ad hoc languages that overfit to the training tasks and partners, at the expense of general properties like interpretability and compositionality (Tucker et al., 2022). In other words, purely utility-driven emergent language might maximize immediate team performance but produce messages that are incomprehensible to outsiders or not generalizable to new contexts. In this work, we argue that incorporating social learning can improve the trade-off between language informativeness and utility under the framework of information bottleneck theory (Tishby et al., 2000).

In this work, we propose to employ language-grounded Theory of Mind (LG-ToM) reasoning as a novel intrinsic reward signal for multi-agent communication. The idea is to equip agents with the ability to infer the mental states of others (intentions, knowledge) using natural language representations, and to use this capability to guide their communication and coordination. By leveraging the priors and commonsense reasoning of large language models (LLMs), agents can simulate a Theory of Mind for their peers and compute a social learning reward based on how their messages change other agents' belief states. This approach provides a dense learning signal that allows agents to learn what information is important to share in order to help the team, without needing explicit domain-dependent reward shaping. The main contributions of this work are three-fold:

- We introduce a novel framework, LG-TOM, that uses language-grounded Theory of Mind to generate social learning rewards for multi-agent reinforcement learning with communication.

- We demonstrate that leveraging priors from Large Language Models for belief estimation allows for effective social learning without access to the privileged information of other agents.

- We show empirically that our approach leads to more effective and efficient communication protocols compared to state-of-the-art baselines in challenging collaborative tasks.

## 2 RELATED WORK

### 2.1 SOCIAL LEARNING IN MULTI-AGENT SYSTEMS

Social learning allows agents to acquire knowledge and skills by observing or interacting with others, which can help them discover more complex policies and generalize better to novel environments (Ndousse et al., 2021). This can involve agents shadowing the behaviors of experts to learn how to act and communicate in collaborative tasks (Karten et al., 2023a). Jaques et al. (2019) proposed giving an agent an intrinsic "social influence" reward when its actions significantly affect other agents' behavior. This encourages agents to communicate or act in ways that impact their teammates, effectively learning from each other. Our work shares this spirit of using social influence as auxiliary rewards, but we craft the reward via Theory-of-Mind reasoning in which agents are rewarded for communication that shapes others' beliefs, not just their immediate actions. Additionally, the modeling of other agents is updated based on the language priors provided by LLMs instead of relying on partial observation of others. This provides MARL agents with higher-level social reasoning capabilities grounded in pretrained language models.

Recent research has also begun to explore the use of LLMs as an external knowledge base to provide social learning cues. For example, Lin et al. (2024) proposed learning the reward function for MARL feedback instead of relying on extensive reward shaping. LLMs can also be used to guide the communication of MARL agents to align their emergent language with human-interpretable concepts, facilitating better human-agent collaboration (Li et al., 2024). In our approach, LLMs are used to provide language groundings for learning the belief estimation network during training based on their social reasoning capabilities (Kosinski, 2023; Li et al., 2023).

## 2.2 LLMs as Social Reasoners

Recent studies have shown that state-of-the-art large language models like GPT-4 can solve classic false-belief tasks, performing comparably to 9-year-old children (Kosinski, 2023). This suggests that an implicit ToM-like capability may have emerged as a byproduct of learning to model the statistical patterns of human language. However, the research community has also raised concerns about the robustness of these abilities. Follow-up work has demonstrated that LLMs may often fail on trivial alterations to the standard ToM tests, suggesting their success may stem from sophisticated pattern matching on familiar problem structures rather than formal social reasoning (Ullman, 2023; Sap et al., 2023). This has led to the development of more challenging benchmarks to evaluate higher-order ToM in dynamic, interactive teamwork scenarios, where current models still fall short of human performance (Kim et al., 2023; Li et al., 2023).

This ongoing debate informs our approach. We propose that simply using an LLM as a zero-shot, off-the-shelf social reasoner is likely to be insufficient for robust performance in complex MARL environments. Instead, the rich social priors embedded within an LLM should be used to initialize and structure a ToM model that is then continuously refined through environmental interaction and feedback. This is the core principle behind our **LG-TOM** framework, which aims to combine the scalable, task-agnostic knowledge of LLMs with the grounding and adaptation capabilities of reinforcement learning.

## 2.3 Multi-agent Communication

There is a rich literature on enabling agents to develop their own communication protocols in MARL. Early frameworks such as DIAL (Foerster et al., 2016) and CommNet (Sukhbaatar et al., 2016) demonstrated that agents can learn to exchange messages through differentiable communication channels during end-to-end training. Subsequent approaches introduced structured or targeted communication to improve efficiency via gating or attention mechanisms, e.g., IC3Net (Singh et al., 2018), IMMAC (Sun et al., 2021), TarMac (Das et al., 2019), MAGIC (Niu et al., 2021). More recently, researchers have tried to ground communication in agents' observations via autoencoders (Lin et al., 2021) or contrastive learning (Lo et al., 2023). However, these MARL communication frameworks use unconstrained continuous message channels where agents can send dense vectors or gradients to each other, essentially turning a multi-agent system into one large differentiable model (Lazaridou & Baroni, 2020). In this work, we focus on a communication channel with a limited number of discrete tokens. This constraint creates pressure for emergent communication to be efficient, meaningful, and potentially generalizable to foundation language models.

Previous work has also explored iterative reasoning in multi-agent systems. InfoPG (Konan et al., 2022) assumes agents share action probabilities to facilitate iterative reasoning to optimize long-term influence. In contrast, our agents utilize a separate communication head to output high-dimensional vectors, allowing for the transmission of richer semantic information. Similarly, while PR2-AC (Wen et al., 2019) provides a theoretical framework for probabilistic recursive reasoning, our work extends this by introducing a semantically meaningful belief and communication space grounded in natural language.

## 2.4 Theory of Mind Guided Communication

Theory of Mind modeling has been widely applied to improve coordination in multi-agent systems. By explicitly modeling the beliefs or intents of other agents, a team can achieve more effective collaboration (Oguntola et al., 2023; Qi & Zhu, 2018; Fuchs et al., 2021). ToM2C introduced an explicit ToM module for agents to predict others' goals and decide with whom to share information (Wang et al., 2021). Similarly, MAIC allows agents to send incentive messages to specific teammates based on an internal model of those teammates (Yuan et al., 2022). Jaques et al. (2019) also introduced a Model of Other Agents (MOA) to calculate social influence rewards in decentralized training settings. These methods are mostly limited by their reliance on ground-truth access to other agents' internal states or predefined belief representations grounded in task environments. For example, in Oguntola et al. (2023), belief is defined as the index of the target landmark, and the ground-truth beliefs of all agents are accessible during centralized training. In contrast, LG-ToM attempts to utilize natural language as a task-agnostic belief representation and distills the social reasoning capabilities of LLMs to learn a ToM model without requiring privileged information from other agents.

## 3 PRELIMINARY

### 3.1 DEC-POMDP WITH COMMUNICATION

We formulate the problem as a decentralized partially observable Markov Decision Process with communication (Oliehoek et al., 2016), specified by the tuple $(\mathcal{I}, \mathcal{S}, \mathcal{A}, \mathcal{C}, \mathcal{T}, \Omega, \mathcal{O}, \mathcal{R}, \gamma)$. Here, $\mathcal{I}$ denotes the finite index set of $n$ agents; $s \in \mathcal{S}$ represents the global state; $\mathcal{A}$ is the joint action space; and $\mathcal{C}$ is the joint space of communication messages. The transition function $\mathcal{T} : \mathcal{S} \times \mathcal{A} \rightarrow \mathcal{S}$ maps the current state $s_t$ to the next state $s_{t+1}$ based on the joint action. Because agents have partial observability of the environment, each agent receives a local observation $o^i \in \Omega$ drawn by the observation function $\mathcal{O} : \mathcal{S} \times \mathcal{C} \times \mathcal{I} \rightarrow \Omega$. The reward function is $\mathcal{R} : \mathcal{S} \times \mathcal{A} \rightarrow \mathbb{R}$, and $\gamma \in [0, 1)$ is the discount factor. At each time step $t$, after observing a partial view of the task state $s_t$ together with the communication messages broadcast at the previous step $c_{t-1}$, agent $i$ selects an action $a_t^i$ and issues a message $c_t^i$. The agent then receives an individual reward $r_t^i \in \mathcal{R}(s_t, a_t)$ from the environment. Unless other wise notice we focus on fully cooperative teams, where learning aims to maximize the total expected discounted return across agents:

$$\max_{\pi^i : \Omega \rightarrow \mathcal{A} \times \mathcal{C}} \mathbb{E} \left[ \sum_{t \in T} \sum_{i \in \mathcal{I}} \gamma^t \mathcal{R}(s_t^i, a_t^i) \,\big|\, a_t^i \sim \pi^i, \, o_t^i \sim \Omega \right] \tag{1}$$

### 3.2 THEORY OF MIND INTRINSIC REWARD

We define an intermediate representation of agent belief as $b \in \mathcal{B}$. Each agent $k$ maintains its own belief $b_t^k$ to generate action $a_t^k$ and communication $c_t^k$ at timestamp $t$. $b_t^k$ is sequentially updated based on the input observation $a_t^k$ and communication vector $C_t$ received from other agents. From a Theory of Mind (ToM) perspective, an agent's ability to reason about how its communication messages influence the internal mental states of others is critical for effective coordination. We borrow the definition of social influence from previous literature (Jaques et al., 2019) to compute the intrinsic reward based on Theory of Mind reasoning. Consider two agents, $k$ and $j$, where agent $k$ models the belief of agent $j$ as a distribution over possible hidden states, denoted $p(b_t^j \mid \cdot)$. Specifically, agent $k$ maintains a ToM model that estimates how agent $j$'s belief state would change in response to different communication actions $c_t^k$.

When agent $k$ communicates using $c_t^k$, it predicts that agent $j$ will update their belief to $p(b_t^j \mid c_t^k, o_t^k)$, conditioned on the communication and agent $k$'s observation. To evaluate the influence of this communication, agent $k$ performs a counterfactual reasoning: it considers a distribution over alternative messages $\tilde{c}_t^k$, computes the resulting belief states $p(b_t^j \mid \tilde{c}_t^k, o_t^k)$, and averages them to estimate agent $j$'s marginal belief:

$$p(b_t^j \mid o_t^k) = \sum_{\tilde{c}_t^k} p(b_t^j \mid \tilde{c}_t^k, o_t^k) \, p(\tilde{c}_t^k \mid o_t^k).$$

This marginal belief reflects what agent $j$ would believe in the absence of any specific message from agent $k$. The discrepancy between the conditional and marginal beliefs, $Dis\left[ p(b_t^j \mid c_t^k, o_t^k) \,\|\, p(b_t^j \mid o_t^k) \right]$, quantifies the extent to which agent $k$'s communication shapes agent $j$'s internal model of the world. By summing over all other agents $j \neq k$, agent $k$ obtains an intrinsic reward that reflects its causal influence on others' mental states:

$$r_t^{int,k} = \sum_{j \neq k} Dis\left[ p(b_t^j \mid c_t^k, o_t^k) \,\|\, p(b_t^j \mid o_t^k) \right].$$

Note the above Theory of Mind reasoning is conducted from the ego-centric perspective of agent $k$. This reward encourages agent $k$ to communicate in ways that meaningfully impact the beliefs of others. Essentially, this is asking agent $k$ a retrospective question: "How would other agents' belief state change if I had communicated differently in this situation?". We will further explore the impact of such a belief estimation model on Theory of Mind guided multi-agent communication in later sections.

It is worth pointing out that the framework assumes a discrete communication channel where gradient information is not shared between agents. Specifically, the generation of the communication vector is treated as a discrete stochastic action which inherently blocks gradient flow. This distinguishes our approach from differentiable frameworks that allow backpropagation from receiver to sender. This is also why the intrinsic reward is essential since it provides the necessary social learning signal to optimize the communication policy in the absence of direct gradient feedback from other agents.

Additionally, although the belief state is modeled as a dense vector (word embeddings), we can still perform causal analysis by measuring the probability distribution of communication tokens and calculating the mutual information between Agent $k$'s message $c_t^k$ and Agent $j$'s action distribution $a_{t+1}^j$. As described later in Section 5.3, this process allows us to quantify the impact of communication on other agents' action distribution.

## 4 METHOD

### 4.1 MODEL STRUCTURE

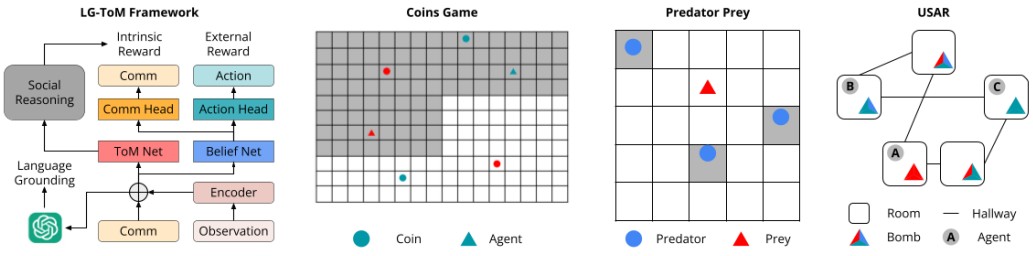

Figure 1: Illustrations of our proposed model structure and benchmark environments.

The proposed model structure of our online RL framework, **LG-ToM**, is shown in Fig. 1. Each agent is equipped with a fully connected neural network as the observation encoder to encode the individual observation $o_t^k$ into high-dimensional vector. This feature vector is then concatenated with the receiving communication vector from the last timestamp $C_{t-1}$ to form the input feature vector to belief network. The belief network is a recurrent neural network taking input vector and hidden state from previous timestamp to form the belief state output $b_t^k$. The ToM model is a dense neural network that takes ego Agent $k$'s observation $o_t^k$ and the communication message from other agents $c_{t-1}^j$ (where $j \neq k$) as input. The output is compared with the ground truth belief provided by LLM agents to calculate the supervised learning loss. The parameters of the ToM model are updated via a combination of reinforcement learning and supervised learning losses. Two separate heads are used to output the communication $c_t^k$ and action $a_t^k$ selection for agent $k$. The agent selects a discrete communication token via Gumbel Softmax (Jang et al., 2016) to form a one-hot vector. This vector is then mapped through a learnable prototype matrix $T \in \mathbb{R}^{z \times c}$, so the transmitted message is the embedding of one of $z$ learned communication tokens. Additionally, a gating function is used to constrain the communication frequency.

The Theory of Mind model is updated with supervised learning in which the output belief prediction is compared to the ground truth of other agent' belief state. During policy rollout, the RL agent receives external guidance from LLMs in an online fashion by putting LLM agents into the same situation. In practice, we utilize an offline dataset $D$ solely to mitigate the latency of querying LLMs online. The external offline dataset $D$ is generated by embodied Large Language Model agents as introduced in Section 4.1.1. During training, we sample a reference belief description from the dataset based on other agent's observation and action, $b_{gt} = D(o_t^j, a_t^j)$, representing how a human (LLM) would describe their belief state in the same situation. We then calculate the cosine similarity between the belief prediction $\mathbf{b}_t^j$ and the word embedding of the reference belief $b_{gt}$ as part of the supervised learning loss.

The Theory of Mind model and communication head is updated via a combined loss of reinforcement learning loss and supervised learning loss. This encourage the agent to learn communication in order to optimize for two learning objectives: 1) maximizing environmental reward by determining useful information to share with other agents based on the policy loss gradient, and 2) maximizing the alignment between the predicted belief with the given ground truth based on the supervised learning loss. The total loss function is formulated as follows:

$$L = L_{RL} + \lambda \sum_{t \in T} \sum_{i \in \mathcal{I}} \left[ 1 - cos(\mathbf{b}_t^i, b_{gt}) \right] \tag{2}$$

Intrinsic reward $r^{int}$ from counterfactual reasoning and external reward $r^{ext}$ from the environment are used jointly for the reinforcement learning. Based on empirical results, intrinsic reward is used to update the communication head and external reward is used to update the action head as shown in Fig. 1. Empirical results suggest that this setup provide the optimal balance between social learning and environmental reinforcement Jaques et al. (2019); Li et al. (2023; 2024). Since the MARL agents optimize their own policies for task rewards while only aligning their internal representations with belief annotations, they utilize the semantic grounding without inheriting the LLM's suboptimal behaviors. Additionally, we approximate beliefs using neural belief encoders, producing continuous embedding vectors $\mathbf{b}_t^j \in \mathcal{R}$ rather than explicit distributions. These vectors are compared using cosine similarity as a proxy for belief change. Therefore, the intrinsic reward is defined as follows in practice:

$$r_t^{int,k} = \sum_{j \neq k} \left[ 1 - \cos \left( \mathbf{b}_t^j(c_t^k, o_t^k), \ \mathbf{b}_t^j(o_t^k) \right) \right] \tag{3}$$

### 4.1.1 LANGUAGE GROUNDED BELIEF ESTIMATION

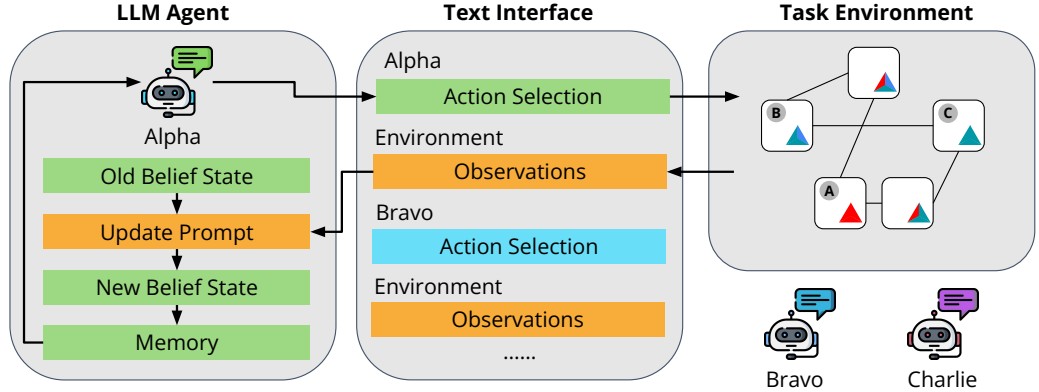

Figure 2: Illustration of the pipeline we use to collect language grounded belief state data. Multiple LLM-based agents interact with the target task environment through a text-based interface. The interface maps agents' language outputs to abstract actions and translates environment feedback into observations. At each timestamp, agents receive observations, update internal beliefs, and produce both next actions and messages to interact with the environment.

To obtain an estimation of the ground truth belief state of teammate agents in a collaborative task, we employ Large Language Models (LLMs) as the embodied agents and the annotators of their own beliefs as shown in 2. Specifically we duplicate the target multi-agent environment in the textual space and allow LLM agents to interact with each other, where they receive natural language observations and communicate with teammates to achieve a common goal. Since agents cannot directly observe each other's actions, they must infer their teammates' mental state from communication in

order to coordinate tightly. To facilitate teamwork and make the ToM reasoning process explicit, we engineer a prompt that instructs each agent to maintain and update a textual representation of its current belief state. At each round, the agent updates this structured text based on new environmental feedback and messages from its teammates. This updated belief state is then carried forward in the agent's prompt history, serving as a dynamic memory for future planning and action selection. Previous research has shown this method externalizes the agent's reasoning process thus improves their collaborative performance as well as ToM capability without requiring explicit update rules. The resulting sequence of explicit belief states serves as our ground truth data for theory of mind analysis. This methodology allows us to collect a offline dataset of interactions where an agent's internal beliefs are clearly articulated at every step of the collaborative task. Further implementation details on the text-game interface and LLM agent setup are available in the appendix.

## 4.2 Experiment setups

We evaluate our proposed method in two multi-agent collaborative tasks. The first, Predator-Prey (Singh et al., 2018), is a standard benchmark in communication-based multi-agent reinforcement learning (comm-MARL). In this gridworld environment, a team of predators with a limited field of view must search for a stationary prey. This task primarily requires agents to share their partial observations to form a complete understanding of the environment's state. The second and more demanding environment, Urban Search & Rescue (USAR) (Oguntola et al., 2021), simulates a team of specialists searching for and defusing bombs in an unknown environment with limited observability. Because each specialist has the unique capability of defusing bombs in different stages coded by colors, this task involves heterogeneous agents and temporal dependencies between actions. To succeed, agents must engage in more complex communication, sharing not only their observations but also their intentions and requests for effective coordination. We also evaluated our method in SocialJax (specifically the Coin Game) (Guo et al., 2025). The environment features sequential social dilemmas where agents must trade off individual and collective interests. This task is significantly more challenging due to a larger observation/action space, a mixed strategy space (cooperative and competitive), and a fully decentralized setup where agents lack access to others' internal states or rewards. Illustrations of both environments are shown in Figure 1, and further details are provided in the Appendix.

To benchmark our proposed **LG-ToM** framework, we compare it against four previous methods including prototype-based discrete communication (*Proto*) (Tucker et al., 2021), autoencoder communication (*Autoencoder*) (Lin et al., 2021), language grounded communication (*LangGround*) (Li et al., 2024), social influence reward (*Social Influence*) (Jaques et al., 2019), and InfoPG (Konan et al., 2022) Unless otherwise noted, all models employ a standard Multi-Agent Reinforcement Learning (MARL) architecture. Each agent uses an observation encoder, a recurrent belief network (GRU) to maintain a belief state $b_t^k$, and separate output heads for actions and communication. In Predator-Prey and USAR, all models are trained with REINFORCE (Williams, 1992) using the centralized training and decentralized execution (CTDE) paradigm with shared parameters between agents. While in SocialJax, all agents have separate policy networks and are trained with IPPO De Witt et al. (2020) with individual rewards in fully decentralized manner.

## 5 Results

### 5.1 Task Performance

As shown in Table 1, our method outperforms the benchmark baselines on, Predator–Prey (v0) USAR , and

By contrast, in the easier variant where predators have a larger field of view, a ceiling effect is observed as all methods perform near-optimally. This outcome aligns with Information Bottleneck theory: when communication bandwidth is ample for the task's demands, the need for sophisticated strategies to optimize informativeness is reduced, diminishing the advantage of social learning.

Table 1: Task performance measured by completion rate and coins collected (mean ± SE).

| Method | PP v1 | PP v0 | USAR | Coin Game |
|---|---|---|---|---|
| **LG-ToM (Ours)** | $0.96 \pm 0.04$ | $\mathbf{0.56} \pm 0.01$ | $\mathbf{0.93} \pm 0.04$ | $\mathbf{98.38} \pm \mathbf{2.38}$ |
| Proto | $0.82 \pm 0.13$ | $0.40 \pm 0.05$ | $0.67 \pm 0.27$ | $96.25 \pm 3.02$ |
| Autoencoder | $\mathbf{1.00} \pm 0.00$ | $0.38 \pm 0.16$ | $0.71 \pm 0.24$ | $78.70 \pm 23.72$ |
| LangGround | $0.95 \pm 0.03$ | $0.38 \pm 0.11$ | $0.68 \pm 0.22$ | $96.68 \pm 0.77$ |
| Social Influence | $0.90 \pm 0.08$ | $0.47 \pm 0.01$ | $0.53 \pm 0.23$ | $93.41 \pm 1.27$ |
| InfoPG | N/A | N/A | N/A | $92.83 \pm 2.48$ |

## 5.2 ABLATION EXPERIMENT

We designed six ablation conditions to isolate the effects of intrinsic rewards and different Theory of Mind (ToM) supervisions:

1. **Baseline:** A standard MARL architecture trained with only the external environmental reward.

2. **Intrinsic:** The Baseline model augmented with a counterfactual intrinsic reward, calculated using direct access to other agents' ground-truth beliefs during training.

3. **ToM:** The Baseline model with a ToM network trained online via a supervised loss on teammates' ground-truth internal belief states.

4. **Ground_ToM:** Similar to the `ToM` condition, but the ToM network is supervised using an offline, LLM-generated dataset ($D$) of natural language belief descriptions.

5. **Intrinsic + ToM:** Combines the intrinsic reward and the ToM network. The reward is calculated using the *predicted beliefs* from the agent's own ToM model.

6. **Intrinsic + Ground_ToM (Full model):** Our proposed **LG-ToM** model, which combines the intrinsic reward (based on predicted beliefs) with a ToM model trained using offline, language-grounded supervision.

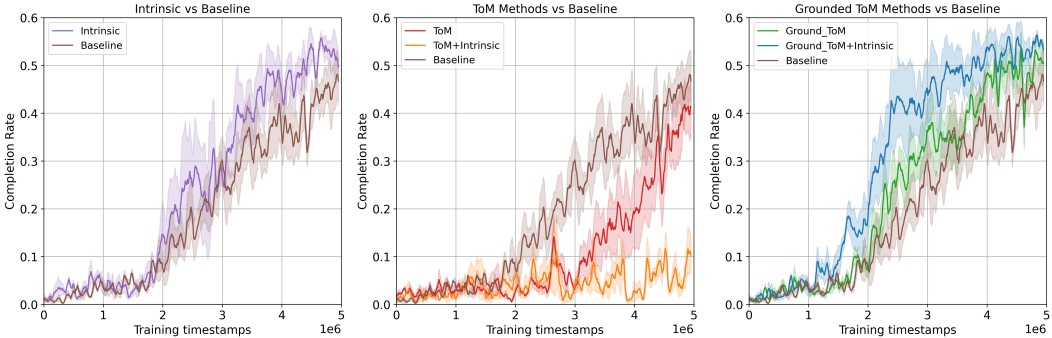

Figure 3: Ablation experiment results in Predatory Prey (vision = 0) environment, separated by conditions. The left subplot compares the Intrinsic condition with baseline, indicating the benefit of introducing dense social learning signals. The middle subplot illustrates that solely relying on internal ToM model may lead to unstable learning outcomes. The right subplot compares the conditions with language-grounded against the baseline.

Primary findings of the ablation experiment are presented in Figure 3. The `Intrinsic` condition outperforms the `Baseline` (left subplot), as the dense intrinsic reward provides a social learning signal by evaluating a message's impact on a teammate's beliefs. This encourages informative communication but relies on direct access to ground-truth belief states, which may be impractical in decentralized settings.

Counterintuitively, adding an online-trained ToM network (`ToM` and `Intrinsic + ToM`) degrades performance below the `Baseline` (middle subplot). This failure stems from two issues: conflicting gradients between the supervised ToM loss and the RL objective, and the "moving target" problem where the ToM network learns from teammates whose own belief networks are simultaneously updating. This instability creates inaccurate belief predictions, which in turn generate a noisy intrinsic reward in the `Intrinsic + ToM` case, leading to accumulation of errors and the worst overall performance.

The language grounded approach yields significant benefits (right subplot). By training the ToM on a stable offline dataset of LLM-generated belief descriptions (`Ground_ToM`), we eliminate the "moving target" instability, resulting in better coordination. Our full model `Intrinsic + Ground_ToM` achieves the best performance. It develops a reliable ToM via language grounded supervision and uses this model to generate intrinsic rewards for communication. This combination encourages agents to accurately model teammates and send messages that effectively shape their beliefs, leading to the most robust and efficient communication protocol.

### 5.3 INFORMATION-THEORETIC ANALYSIS

Table 2: Conditional Mutual Information (CMI) of communication (mean $\pm$ SE).

| **Method** | $I(c_t^k; a_{t+1}^j \mid o_t^j)$ | $I(c_t^k; o_{t-1}^j \mid o_t^j)$ |
| --- | --- | --- |
| LG-ToM | $\mathbf{0.18} \pm 0.01$ | $\mathbf{0.34} \pm 0.02$ |
| Proto | $0.14 \pm 0.03$ | $0.29 \pm 0.05$ |
| Autoencoder | $0.16 \pm 0.01$ | $0.26 \pm 0.02$ |
| LangGround | $0.09 \pm 0.02$ | $0.15 \pm 0.03$ |
| Intrinsic | $0.08 \pm 0.04$ | $0.11 \pm 0.05$ |

To further evaluate the learned communication protocols, we quantify the conditional mutual information (CMI) between an ego agent $k$'s communication token $c_t^k$ and teammate agent $j$'s variable (e.g., observation $o_t^j$, action $a_t^j$, or belief $b_t^j$) conditioned on agent $k$'s own context. This measurement captures the residual dependence between messages and the teammate after accounting for the ego agent's own information. A higher CMI suggests the communication protocol is more adaptive to teammate-related factors. For a causal interpretation, we also compute time-lagged variants, e.g., $I(c_t^k; a_{t+1}^j \mid o_t^j)$, to quantify the impact of message at $t$ on the teammate's action at $t + 1$. Table 2 shows the conditional mutual information measurements in Predator Prey environment. The communication protocols learned by **LG-ToM** yield the highest mutual information for both action-conditioned and observation-conditioned measures, compared to other methods. This helps explain its superior team performance: **LG-ToM** agents employ a more teammate-aware communication strategy, which leads to more effective collaboration.

Additionally, we visualize how communication varies with teammate observation using a heatmap as shown in Figure 4. Rows denote communication tokens, columns denote teammate observations, and color encodes token frequency within a fixed ego context. The top subplot (i.e. **LG-ToM**) shows a more diverse frequency distribution where agents send out different tokens in response to their teammate's state. While the bottom subplot shows a uniform distribution of communication token across teammate state, indicating an ego-centric communication strategy which may not be ideal for tight coordination.

## 6 CONCLUSION

Our experimental results offer key insights into the challenges and opportunities of integrating Theory of Mind (ToM) into multi-agent reinforcement learning with communication. The key finding is that how the ToM module is supervised is critical to its effectiveness. A naively implemented ToM trained online using the concurrent belief states of other agents harmed performance. As shown in our ablation study, this failure stems from the "moving target" problem and conflicting gradients, which generate inaccurate belief predictions and noisy intrinsic rewards. This result is a strong caution that simply adding social reasoning modules is insufficient without a stable learning foundation.

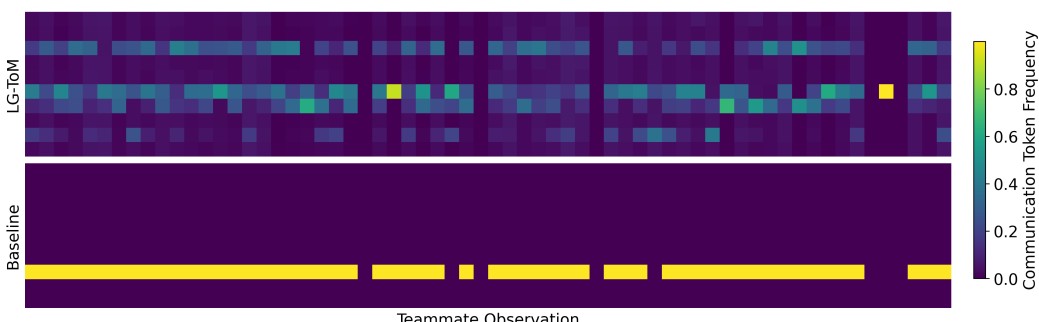

Figure 4: Conditional message heatmaps $p(C|E, T)$. Rows are communication tokens, columns are teammate observations, and color shows token frequency in the given ego context. Vertical variation across columns signals teammate-aware communication, whereas uniform rows indicate ego-dominant messaging.

The success of our language grounded Theory of Mind approach, **LG-ToM**, offers a promising solution. By supervising the ToM model with a static, offline dataset generated by a Large Language Model (LLM), we anchor agents' understanding of belief states, providing a stable and sensible foundation for social reasoning. This decouples the challenging task of belief modeling from the online reinforcement learning process, allowing agents to build reliable models of their teammates. Our information theoretic analysis provides a clear mechanism for this success: **LG-ToM** learns a more teammate-aware communication protocol guided by the social learning signal.

This work points toward a paradigm where large, pre-trained models provide the semantic grounding required for smaller, specialized agents to efficiently learn complex social behaviors. The method could extend beyond communication to other areas of multi-agent coordination, such as joint planning or role allocation, where shared understanding of intent and belief is crucial. Additionally, most foundation models are trained within reinforcement learning frameworks that optimize rigid reward functions derived from limited human feedback or narrowly structured tasks (e.g., mathematics, coding). Social learning remains underexplored as a means to enhance the reasoning capabilities of foundation models, especially in multi-agent settings.

## 7 REPRODUCIBILITY STATEMENT

The full source code for the proposed reinforcement learning algorithm will be made publicly available upon paper acceptance. Details of LLM-based data collection are provided in the appendix. If the commercial APIs used become unavailable, the raw LLM-generated data can be shared by the authors upon request. Implementation details for training and evaluation are given in the appendix, and the model architecture is specified in Section 4.1 of the main text.

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

## A APPENDIX

### A.1 IMPLEMENTATION DETAILS

#### A.1.1 PREDATOR PREY

In this task, $n$ predators with vision range $v$ search for a stationary prey on an $x \times x$ grid. A predator receives a positive reward upon reaching the prey; an episode terminates when all predators reach the prey or the step limit $T$ is exceeded. Initial predator and prey locations are randomized. At each timestep, a predator observes a $v \times v$ window centered on its position and chooses a movement action. Because observations are partial, predators must communicate to coordinate efficient exploration. We use map sizes $x = 5$ with 3 predators and 1 prey. We vary vision $v \in \{0, 1\}$ to obtain task variants; when $v=0$, predators cannot see the prey until occupying the same cell. The maximum episode length is 20.

#### A.1.2 USAR

USAR simulates cooperative search-and-rescue. Three agents (Alpha, Bravo, Charlie) must locate and defuse color-coded bombs in an unexplored environment. Each bomb has an $m$-color phase sequence revealed only upon inspection. Agents start with different wire cutters and must execute the correct sequence to defuse a bomb. The environment is a graph with $n$ rooms (nodes) connected by hallways (edges). At each timestep, an agent may move to any room, inspect the local bomb sequence, or use one of the $m$ cutters; the action-space size is $n+m+1$. Observations include only the current room's contents and agent status. The team earns $10x$ points for defusing an $x$-phase bomb. Episodes end when all bombs are defused or the time limit is reached. Because capabilities and observations are complementary (each agent carries only a subset of cutters), effective communication is required for timely information sharing and synchronization.

We instantiate USAR with $n=5$ rooms and five bombs (two single-phase, two double-phase, one triple-phase). Phases use $m=3$ colors. Each of the three agents spawns with two distinct cutters, necessitating collaboration on multi-phase bombs. Rewards are 10 points per completed phase (maximum 90 per episode). The episode limit is 100 steps.

### A.1.3  SocialJax (Coin Game)

The Coins environment, originally introduced in (Lerer & Peysakhovich, 2017), features two agents navigating a room to collect colored coins. Coins appear periodically at random locations on a $16 \times 11$ grid. Each generated coin has a $50\%$ probability of matching the first agent's color and a $50\%$ probability of matching the second agent's color. Agents possess a limited field of view of $5 \times 5$. An agent receives a reward of $+1$ for collecting any coin. However, a significant social dilemma arises from the penalty structure: if an agent collects a coin matching the other player's color, that player incurs a penalty of $-2$. Each empty tile has a coin respawn probability of $p = 0.0005$ per step for each color type. This environment serves as a benchmark for analyzing social dilemmas, as agents can choose to maximize their own reward indiscriminately or cooperate by avoiding the other agent's coins. Consequently, we evaluate cooperation by measuring the number of same-color coins collected by each agent.

### A.1.4  Text Game Interface

Predator Prey and USAR are implemented for MARL using the Gym API. To enable LLM interaction, we introduce a rule-based text interface per task. At each timestep, LLM agents act sequentially: they receive a natural language observation, output an action in natural language, and optionally broadcast a message that is appended to the next round's observation for all teammates. LLM agents receive the same information as MARL agents, limited to each agent's partial view.

The interface converts game state into text and maps agent replies back to valid actions. It extracts state features (round index, team score, action feedback, visible objects, and teammate messages) and fills predefined templates to produce structured observations. Action decoding uses keyword matching: models are instructed to respond with specific keywords and formats, which the interface maps to environment actions. Invalid or ambiguous replies trigger informative, rule-based error messages to guide correction. For example, attempting to inspect a non-existent bomb yields: "There is no bomb in Room X for you to inspect."

### A.1.5  Embodied LLM Agents

Large language models (LLMs) are prompted to act in team-based environments. We follow the pipeline of (Li et al., 2023), augmenting agents with explicit belief states and a communication channel to improve collaboration. Each agent maintains a memory of its own observations and teammates' messages. Our prompts specify only general task rules and intentionally avoid prescribing coordination or communication strategies, minimizing prompt-engineering bias and supporting transfer across environments. For belief–ground-truth collection and ad hoc teamwork, we use the OpenAI API with `gpt-4-0125-preview` and set the temperature to 0 for deterministic outputs.

### A.1.6  Language Grounding Belief

To obtain language-grounded beliefs during teamwork, we collect expert trajectories from embodied GPT-4 agents in interactive scenarios. Consistent with our preliminary findings, these agents produce reasonable actions and communications and can explicitly report their beliefs. In $USAR$, we collected 50 episodes resulting in 2,550 pairs of (observation, action) and belief annotations. For the predator-prey environments, we collected 1,893 pairs for $pp_{v0}$ and 2,493 pairs for $pp_{v1}$. In the $CoinGame$, we collected 19 episodes (3,800 pairs) using a GPT-5.1 model. The token usage of $CoinGame$ alone was approximately 4.37M, costing roughly \$43.70. To align ToM model outputs with these beliefs, we embed each natural-language belief description using OpenAI's `text-embedding-3-large`, yielding high-dimensional vectors for supervised training and evaluation.

## A.2 LLM PROMPTS

## A.3 TASK CONTEXT (USAR)

---

**System Prompt: USAR Environment**

Welcome to our interactive text game! In this game, you'll assume the role of a specialist on a search and rescue team. Alongside two other players, you'll navigate a five-room environment with a mission to defuse five hidden bombs.

**The Map:** Imagine a network of rooms represented by a connected graph where each node corresponds to a room, and the edges between nodes depict hallways. The rooms are numbered 0, 3, 6, 5, and 8. Room 0 is connected to all other rooms. Room 5 shares a hallway with room 6. Room 3 is linked to room 8. And room 8 is also connected with room 6. You can only travel to adjacent, directly connected rooms at each turn.

**The Challenge:** Scattered among these rooms are five bombs, each coded with different phases represented by colors. To defuse them, you'll need to use the correct wire-cutting tools in the correct sequence. There are one-phase, two-phase, and three-phase bombs, needing 1, 2, or 3 color-coded tool applications in sequence to disarm. For instance, a bomb with a red-green phase sequence requires the red tool first, then the green one. Points are awarded based on the number of tools used for defusing a bomb, with each tool use worth 10 points. Your task is to maximize the team score as soon as possible. The challenge is that the bomb locations and sequences are unknown to players at the start.

**Tools:** Each player is equipped with two color-coded wire cutters. As player Alpha, you have red and green tools, player Bravo wields green and blue, and player Charlie possesses blue and red.

**Actions:** Each round, you can opt to do one of the following: 1) Move to an adjacent room, 2) Inspect a bomb's phase sequence in your current room, or 3) Apply your wire cutters to a bomb in the current room.

**Communications:** In addition to selecting an action to take from the above list, you can also send communication message texts to both of your teammates in each round. The message text you sent will be shared with both of your teammates in their observation in the next round.

**Observation:** While you can only see what's in your current room and read text messages from teammates. You'll also be informed of the current round number, team score and the current location of your teammates. Your teammates have the same observability as you. They will not be able to know your action and its consequences unless you explicitly communicate.

To facilitate our interaction, reply your action selection and communication messages in this fixed format: Action selection: Your action. Message to Team: "Your Message". To move to an adjacent room, say: 'Move to Room X'. To inspect the sequence of a bomb in your current room, say: 'Inspect Bomb'. To apply a wire cutter tool, say: 'Apply X Tool'. Remember, your replies must adhere strictly to these rules. Feel free to ask clarifying questions if needed. I'll supply the necessary information as we progress. Are you ready to take on this explosive challenge?

---

## A.4 Social Dilemma Context (Coin Game)

---

**System Prompt: Coin Game Agent**

You are an AI agent in a multi-agent environment with a social dilemma.
**ENVIRONMENT DESCRIPTION:**

- You are Agent {self.agent_id} with {self.team_color} color
- There is another agent in the environment with a different color
- Red coins and green coins spawn randomly throughout the map
- You can move around to collect coins and communicate with the other agent

**MAP AND COORDINATE SYSTEM:**

- Map size: 16 rows × 11 columns grid
- Coordinate system: (x, y) where:
    - Origin (0, 0) is at the SOUTHEAST corner
    - X-axis: increases from South (0) to North (16)
    - Y-axis: increases from East (0) to West (11)
- Your position is given as (x, y) coordinates
- Example: position (8, 5) means 8 steps north from south edge, 5 steps west from east edge

**DIRECTIONS AND ORIENTATION:**

- You have a facing direction: North, East, South, or West
- Direction meanings in the coordinate system:
    - North: facing towards higher x values (increasing x)
    - South: facing towards lower x values (decreasing x)
    - East: facing towards lower y values (decreasing y)
    - West: facing towards higher y values (increasing y)

**FIELD OF VIEW (FOV):**

- Your vision is asymmetric based on your facing direction:
    - Forward: 9 steps in the direction you're facing
    - Backward: 1 step behind you
    - Left/Right: 5 steps on each side
- Objects are described with their absolute (x, y) positions and relative directions

**REWARD STRUCTURE:**

- When you collect a coin matching YOUR color: you receive +1 point
- When you collect a coin matching the OTHER agent's color: you receive +1 point, but the other agent receives -2 points
- The other agent faces the same reward structure

**GOAL:**

- Your goal is to maximize your own score, which means collecting as many coins as possible, while also trying to minimize the penalty from the other agent.

**COMMUNICATION:**

- You can send messages to the other agent
- The other agent can send messages to you
- Messages can be used to negotiate, coordinate, or influence behavior

**AVAILABLE ACTIONS:**

- turn_left: Rotate 90 degrees counterclockwise

---

- turn_right: Rotate 90 degrees clockwise

- left: Move west (strafe)

- right: Move east (strafe)

- up: Move north

- down: Move south

- stay: Stay in place

**OUTPUT FORMAT (must follow exactly):**
BELIEF: [One sentence describing your current understanding of the situation including current position, next goal, and general game strategy.]
ACTION: [One of: turn_left, turn_right, left, right, up, down, stay]
COMMUNICATION: [Your message to the other agent, or "[No message]"]

## A.5 TRAINING AND EVALUATION SETUPS

Agents communicate using discrete tokens embedded in 256-dimensional vectors. The vocabulary size is 10 tokens in Predator Prey and 36 tokens in USAR. All methods share the same model architecture described in Section 4.1. Unless noted, network hidden sizes are 256; the communication head uses a 64-dimensional hidden layer.

We use RMSProp with a learning rate of $1 \times 10^{-3}$ for Predator Prey and $1 \times 10^{-4}$ for USAR. The intrinsic reward coefficient is 0.1, and the supervision loss weight is 1.

All models are trained using the centralized training and decentralized execution (CTDE) paradigm with shared parameters between agents. We use REINFORCE (Williams, 1992) to train the model end-to-end. Training was performed on an AMD Ryzen 7 9800X3D processor. Each method is trained with 3 random seeds for $5 \times 10^6$ environment steps. For evaluation, we run 100 episodes per method per seed and report the mean $\pm$ standard error (SE) across seeds.

## A.6 ABLATION EXPERIMENT

The key differences between the conditions are summarized in Table 3.

Table 3: Summary of Experimental Conditions

| Condition | Intrinsic Reward | ToM Model | ToM Supervision | Counterfactual Source |
|---|---|---|---|---|
| Baseline | × | × | N/A | N/A |
| Intrinsic | ✓ | × | N/A | Ground-Truth Beliefs |
| ToM | × | ✓ | Ground-Truth | N/A |
| Ground_ToM | × | ✓ | Language Grounding | N/A |
| ToM+Intrinsic | ✓ | ✓ | Ground-Truth | Predicted Beliefs |
| **LG-ToM (Ours)** | ✓ | ✓ | Language Grounding | Predicted Beliefs |

## A.7 COMPUTATIONAL COST ANALYSIS

We trained all models for $5 \times 10^7$ timesteps on a single NVIDIA GeForce RTX 5090 GPU with 512 parallel environments. As shown below, the counterfactual reasoning process in LG-ToM and Social Influence introduces a bottleneck ($O(N^2)$ complexity).

## A.8 INTERPRETABILITY AND GENERALIZATION

We conducted a topology analysis to map the language-grounded belief state to the actual agent state. The results demonstrate a high degree of topological similarity: similar agent states map to proximal belief states in the embedding space. This confirms that the belief states learned by

Table 4: Computational Cost comparison.

| Method | Runtime (Minutes) |
|---|---|
| **LG-ToM (Ours)** | 10.56 |
| Social Influence | 10.60 |
| Autoencoder | 6.65 |
| LangGround | 4.70 |
| Proto | 4.70 |
| InfoPG | 4.13 |

LG-ToM capture dynamic changes in team knowledge and intents. The semantic nature of our belief representation allows for high data efficiency and generalization to unseen states. As an example, in the Coin Game, despite this offline dataset covering only 40% of the state space, it provided considerable performance gains. Because natural language captures universal semantics, this representation allows for zero-shot transfer to novel domains.

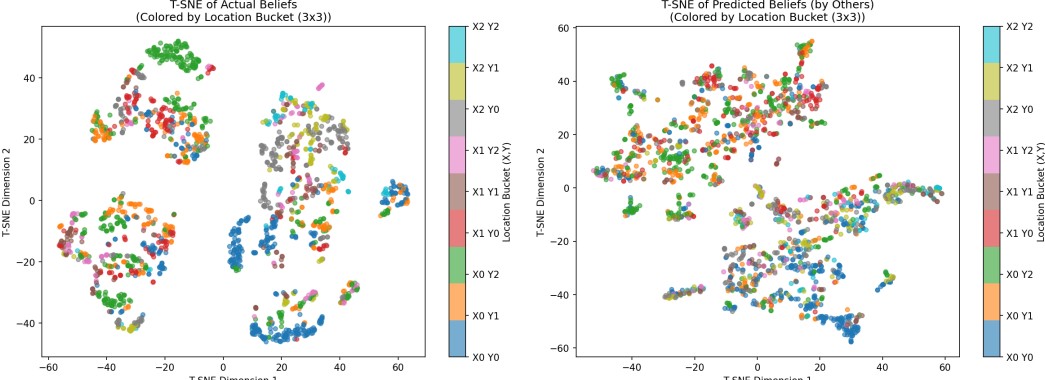

Figure 5: T-SNE visualization of agent's position and their own belief state / the predicted beliefs by other agent's ToM model. The results demonstrate a high degree of topological similarity: similar agent states map to proximal belief states in the embedding space. This confirms that the belief states learned by LG-ToM capture dynamic changes in team knowledge and intents.

### A.9   LLM USAGE

Large language models (LLMs) were used to help search for relevant literature and to polish language in author-written drafts. All LLM-produced text were reviewed and verified by the authors before inclusion.

