# OpenReview forum: "LG-TOM: Language Grounded Theory of Mind Modeling for Multi-agent Communication"
_ICLR.cc/2026/Conference — Submitted to ICLR 2026_

### Official Review · Reviewer_QgbJ · 2025-11-01

**Soundness:** 2
**Presentation:** 2
**Contribution:** 2
**Rating:** 4
**Confidence:** 4

**Summary:**

This paper introduces LG-TOM, a framework for multi-agent communication that incorporates language-grounded Theory of Mind to improve social learning. A belief estimation network, guided by large language model priors, enables agents to infer and influence teammates’ beliefs, while a counterfactual reward promotes belief-shaping communication. Experiments on collaborative tasks show that LG-TOM enhances both performance and communication efficiency, with ablations validating the roles of language grounding and ToM-based rewards.

**Strengths:**

1. Pioneering integration of language-grounded ToM with multi-agent reinforcement learning, introducing a novel social reasoning paradigm.
2. Technically sound framework that effectively addresses training instability via offline LLM supervision, enabling robust belief estimation.
3. Comprehensive experimental validation across collaborative tasks, with rigorous ablation studies and information-theoretic analysis consistently demonstrating effectiveness.

**Weaknesses:**

1. The experimental validation is critically insufficient. The limited scope—only two environments with a handful of agents and a lack of strong, modern baselines—fails to demonstrate the method's generality, scalability, or true competitive advantage.
2. The source of performance gains remains fundamentally ambiguous. The paper does not disentangle the contribution of its novel framework from the mere injection of knowledge from the powerful GPT-4 model, leaving its core intellectual contribution unproven.
3. This work ignores the practical implications of its approach. The substantial computational and financial costs of using a commercial LLM like GPT-4 for data generation are not discussed, creating a significant barrier to adoption and reproducibility for the wider research community.

**Questions:**

1. Could you please provide a detailed description of the architecture and the specific update rules for the ToM network depicted in Figure 1?
2. What information does an agent's state/belief vector b_k explicitly contain, and what is the precise procedure for constructing it from the observation and communication history?
3. Will you release the exact prompts used for the LLM agents to generate the belief-state ground truth? This is critical for reproducibility.
4. Does the proposed method fundamentally rely on a discrete, symbolic observation and action space, as suggested in the abstract? Please clarify its assumptions and limitations regarding the state-action representation.
5. Why were the chosen baseline methods selected, and can you justify that they represent strong, contemporary benchmarks? The performance claims would be more convincing if compared against known state-of-the-art methods in emergent communication.

---

> ### Author Response · Authors · 2025-11-22
> **Response to Reviewer Qgbj (Part 1/2)**
>
> We thank Reviewer Qgbj for acknowledging the novelty and technical soundness of our work. We are pleased that you found our  ablation studies rigorous and information-theoretic analysis helpful. We address your concerns and questions below.
>
> **W1: Extended Evaluation on Social Dilemmas**
> Per your recommendation, we extended our evaluation to a new task domain, **SocialJax [1]**, and included a new baseline, **InfoPG [2]**. The environment features sequential social dilemmas (e.g., the Coin Game) where agents must trade off individual and collective interests. This task is significantly more challenging due to a larger observation/action space, a mixed strategy space (cooperative and competitive), and a fully decentralized setup where agents lack access to others' internal states or rewards.
>
> We selected Advantage-InfoPG to represent state-of-the-art methods in multi-agent communication based on iterative reasoning. As shown below, our method (**LG-ToM**) outperforms all baselines in this complex setting. More details will be added to the Appendix.
>
> | Group | Coins Collected (Mean $\pm$ SE) |
> | :--- | :--- |
> | **LG-ToM (Ours)** | **98.38 $\pm$ 2.38** |
> | LangGround | 96.68 $\pm$ 0.77 |
> | Proto | 96.25 $\pm$ 3.02 |
> | Social | 93.41 $\pm$ 1.27 |
> | InfoPG | 92.83 $\pm$ 2.48 |
> | Autoencoder | 78.70 $\pm$ 23.72 |
>
> **W2: Performance Gain**
> We acknowledge that performance gains could stem from both the expert trajectories and the social learning framework. To isolate the specific contribution of our social learning mechanism, we included the **LangGround** baseline in the original evaluation benchmark. LangGround uses LLM trajectories to ground agent communication but does not utilize social learning. Consequently, the superior performance of LG-ToM over LangGround quantifies the specific benefit of employing Theory of Mind (ToM) as an intrinsic motivation reward in MARL.
>
> **W3: Cost of LLMs**
> We thank the reviewer for highlighting the practical implications of our method. We argue that the cost of LLMs in our pipeline is manageable and justified for two reasons: 1) LLMs are significantly more cost-effective than human annotation. For the Coin Game (19 episodes, 3800 samples), we used a GPT-5.1 model costing approximately $43.70 (4.37M tokens). Collecting equivalent data via human annotation would be financially and environmentally more costly. 2) The semantic nature of the belief representation allows us to use a relatively small offline dataset. In the Coin Game, the offline dataset covers only ~40% of the state space yet provides considerable performance gains. As shown in [3], LLM-based language grounding enables MARL agents to learn a semantic representation space that supports zero-shot generalization to unseen states.
>
> **Q1: ToM Clarification**
> The structure and update process of ToM model is described in Lines 241-254. We will improve the clarity of the method description in the revised paper. The ToM model is a dense neural network that takes ego Agent $k$’s observation $o_t^k$ and the communication message from other agents $c_{t-1}^j$ (where $j \neq k$) as input. The output is compared with the ground truth belief provided by LLM agents to calculate the supervised learning loss. The parameters of the ToM model are updated via a combination of reinforcement learning and supervised learning losses.
>
> **Q2: Belief Vector**
> The belief state of the LG-ToM agent is a dense vector output by the recurrent belief network (as shown in Fig. 1). It is conditioned on the observation and communication input, as well as the hidden state encoding history information.The belief state of LLM agents is a text description maintained via zero-shot reasoning. We did not provide explicit update rules for this belief description but rely on the LLMs’ reasoning capability to update it from observation and communication history. An example belief description in the Coin Game is: *“I’m at (11,9) facing North and will move east toward the red coin at (11,5) while continuing our cooperation that I take red coins and you take green coins so we avoid hurting each other’s scores.”*
>
> **Q3: LLM Prompts**
> We will provide the complete prompts used for LLM data collection in the Appendix.
>
> **References**
> [1] https://arxiv.org/abs/2503.14576
> [2] https://arxiv.org/abs/2201.08484
> [3] https://arxiv.org/abs/2409.17348

---

> ### Author Response · Authors · 2025-11-22
> **Response to Reviewer Qgbj (Part 2/2)**
>
> **Q4: Action/Observation Space**
> We did not imply that our method relies on discrete action and observation spaces in the abstract or other parts of the paper. Our proposed framework is domain-agnostic and can work with any observation/action representation. The LLM trajectory collection functions as long as the task can be executed by expert agents with reasonable belief state annotations. These agents can be replaced by human participants, LLMs, VLMs, or any kind of model depending on the domain. The only assumption we make is that the expert agents are able to provide natural language descriptions of their own belief state, representing task-related information of the situation. Regarding the social learning part, we make no assumptions about the observation-action space because the network structure of the observation embedding and action head is agnostic to the counterfactual reasoning formulation.
>
> **Q5: Baseline Selection**
> The baselines were chosen to represent common methods in constructing messages in emergent communication. **Prototype** methods learn a semantically meaningful communication space for better generalizability and interpretability. **Autoencoder** represents a series of works based on reconstructing individual observations as the means of communication. **LangGround** is one of the few LLM-guided communication methods in the literature. **Social Influence** represents the idea of modeling other agents and the use of intrinsic motivation rewards. Upon the recommendation of reviewers, we added a new baseline, **InfoPG**, to represent recent techniques in iterative reasoning and mutual information. To the best of our knowledge, this selection of baselines represents the state-of-the-art in emergent communication. If you have any better options in mind, please let us know.

---

### Official Review · Reviewer_tBq9 · 2025-11-01

**Soundness:** 2
**Presentation:** 3
**Contribution:** 2
**Rating:** 4
**Confidence:** 3

**Summary:**

LG-TOM introduces a novel framework that combines LLM-based social reasoning with belief model (Theory of Mind) that has been trained with an offline supervised dataset with GPT-4 expert trajectories + textual representations. ToM model and communication policy loss are then updated with a combined standard task reward and an intrinsic social influence reward to promote communication. LG-TOM demonstrates improved performance, sample efficiency, and stability compared to baselines with different communication frameworks.

**Strengths:**

1. Novel emergent behaviors can be achieved using language space embedding and potential for quicker experimentation time, with the use of large-LLM based expert trajectories.
2. I particularly like the choice of baselines that cover a variety of different frameworks for better coverage of agent communication behaviors.
3. Improved stability with the use of an offline trained ToM network using expert trajectories.

**Weaknesses:**

1. Currently lacks evaluation or discussions on generalization capabilities of the agents to novel contexts.
2. Discussions on scalability of the system to larger multi-agents systems has not been discussed in the paper.

**Questions:**

1. How robust is LG-TOM to domain shift? Do you foresee cases where task dynamics do not allow an adaptation?
2. How do you evaluate the belief state shifts correspond to meaningful change in team knowledge or intent? A comparison with actual environment states along with an interpretability analysis would be helpful in understanding the behavior.
3. What is the computation costs associated with LG-TOM as compared to other models? Training steps per second would be a good metric.

---

> ### Author Response · Authors · 2025-11-24
> **Response to Reviewer tBq9**
>
> We thank Reviewer tBq9 for recognizing the strengths of our paper, particularly our use of LLM language groundings to improve and stabilize emergent communication in MARL systems. We also appreciate your validation of our baseline selection. We address your concerns below and will incorporate these revisions into the final paper.
>
> **W1: Generalization**
> We argue that our method demonstrates strong generalizability due to the semantically meaningful belief embedding space with the following evidence:
> 1. **Data Efficiency:** The offline dataset used for training does not cover the entire observation-action space. For instance in the Coin Game, the dataset covers only ~40% of the states while still provides considerable performance gains.
> 2. **Zero-Shot Capabilities:** Previous research [1] has extensively evaluated language-grounded multi-agent communication. It demonstrate that LLM-based groundings enable MARL agents to learn a semantically meaningful representation space, facilitating zero-shot generalization to unseen states. We replicate the similar analysis as shown in Figure 3 in [1] that we map the language-grounded belief state to the actual agent state. The results show a reasonable topological similarity, where close belief states in the embedding space map to similar actual agent states. We believe the learnt belief state in LG-ToM is also semanticly meaningful therefore potentially generalize to unseen states. We will add this new analysis to the revised paper.
>
> **W2: Scalability**
> Our framework is highly flexible and theoretically scales to an arbitrary number of agents, as both the MARL training and LLM data collection pipelines are independent of team size. In practice, a computational bottleneck exists in the counterfactual reasoning process due to $O(N^2)$ complexity. However, optimization techniques such as parallel computation or limiting the reasoning/communication range can effectively mitigate these scalability issues.
>
> **Q1: Domain Shift**
> Domain transfer is a significant challenge in RL, particularly in multi-agent systems where agents must adapt to novel environments and diverse teammate strategies. Although this paper does not focus on domain transfer or meta-RL, our method potentially enables adaptive behaviors via our task-agnostic belief representation (word embeddings). Agents can utilize the trained ToM to predict the impact of communication on teammates' beliefs, provided the beliefs in the new domain can also be expressed in natural language.
>
> **Q2: Interpretability and Semantic Alignment**
> While the primary focus of this work is not interpretability, we conducted an analysis to map the language-grounded belief state to the actual agent state per your request. The results show a reasonable topological similarity, where similar agent states map to proximal belief states in the embedding space. We believe the learnt belief state in LG-ToM captures dynamic changes in team knowledge and intents which potentially leads to better zero-shot generalizability. We will add this new analysis to the revised paper.
>
> **Q3: Computational Cost**
> We trained all models for $5 \times 10^7$ timesteps on a single NVIDIA GeForce RTX 5090 GPU with 512 parallel environments. As shown below, the counterfactual reasoning process in LG-ToM and Social Influence introduces a bottleneck ($O(N^2)$ complexity).
>
> | Method | Runtime (Minutes) |
> | :--- | :--- |
> | LG-ToM (Ours) | 10.56 |
> | Social Influence | 10.60 |
> | Autoencoder | 6.65 |
> | LangGround | 4.70 |
> | Proto | 4.70 |
> | InfoPG | 4.13 |
>
> **References**
> [1] https://arxiv.org/abs/2409.17348

---

### Official Review · Reviewer_8Dmh · 2025-11-02

**Soundness:** 3
**Presentation:** 4
**Contribution:** 3
**Rating:** 6
**Confidence:** 4

**Summary:**

This paper addresses a key challenge in multi-agent reinforcement learning (MARL): learning effective communication protocols in environments with sparse task rewards. The authors propose a novel framework, LG-ToM (Language Grounded Theory of Mind), which equips agents with a social learning mechanism to generate dense, intrinsic rewards.
The core idea is to enable agents to reason about how their messages influence the belief states of their teammates. This "social influence" is then used as an intrinsic reward to encourage more informative communication. The main contribution and novelty lie in how this Theory of Mind (ToM) model is trained. Instead of learning from other agents online—which suffers from instability (the "moving target" problem)—LG-ToM uses a ToM network supervised by a static, offline dataset of belief states. This dataset is generated by Large Language Model (LLM) agents performing the task, grounding the concept of "belief" in natural language descriptions.The primary contributions of the paper are:
1.	The introduction of the LG-ToM framework, which integrates a language-grounded ToM model with an intrinsic social influence reward to guide communication learning.

2.	A novel methodology for training the ToM model using an offline, LLM-generated dataset, which provides a stable learning signal and avoids the need for privileged access to other agents' internal states during training.

3.	Empirical results on two collaborative tasks (Predator-Prey and USAR) demonstrating that LG-ToM outperforms state-of-the-art baselines, leading to more effective and efficient communication protocols that are shown to be more "teammate-aware" through information-theoretic analysis.

**Strengths:**

This is a well-executed and insightful paper that makes a valuable contribution to the field of multi-agent communication. The strengths are evident across all dimensions: originality, quality, clarity, and significance.
	Originality: The paper's primary novelty lies in its creative and effective solution to the "moving target" problem when training Theory of Mind (ToM) models in MARL. While the concepts of social influence rewards and ToM modeling are not new in themselves, the specific methodology of using a stable, offline dataset generated by LLMs to supervise the ToM network is highly original. This approach cleverly circumvents the instabilities of online learning while grounding the agents' abstract belief representations in the rich semantic space of natural language. It presents a novel synthesis of ideas from MARL, intrinsic motivation, and large language models.

1. Quality: The technical quality of the work is high. The methodology is sound and well-motivated. The experimental setup is rigorous, employing standard benchmarks and relevant, state-of-the-art baselines. The ablation study is particularly strong; it systematically dismantles the proposed framework to isolate the contribution of each component. The negative result showing that a naively-trained online ToM model degrades performance is a crucial and convincing finding that strongly justifies the paper's core thesis. Furthermore, the inclusion of an information-theoretic analysis to quantify why the learned communication is better (i.e., more teammate-aware) adds a layer of depth that goes beyond simply reporting task-level scores.

2. Clarity: The paper is exceptionally well-written and easy to follow. The authors do an excellent job of motivating the problem, clearly situating their work within the existing literature, and explaining their proposed method step-by-step. Figures 1 and 2 are clear and provide an excellent overview of the model architecture and data generation pipeline. The narrative of the results, particularly in the ablation section, is compelling and effectively guides the reader through the authors' reasoning and conclusions.

3. Significance: The paper addresses a significant and long-standing challenge in MARL: designing learning signals that promote effective coordination. The findings have important implications. Firstly, it serves as a cautionary tale, demonstrating that simply adding a social reasoning module is not a panacea and can be detrimental if not trained properly. Secondly, and more importantly, it proposes a powerful and promising new paradigm: using large, pre-trained models to provide the semantic grounding and stable supervision needed for smaller, specialized agents to learn complex social behaviors. This work could inspire a new line of research into how the world knowledge of LLMs can be "distilled" to bootstrap more robust and efficient learning in multi-agent systems.

**Weaknesses:**

While the paper is strong, there are several areas where it could be improved. The weaknesses are primarily related to the scalability of the proposed method and the scope of the experimental validation.

1. Dependency on LLM-based Data Generation: The entire method hinges on the costly, environment-specific process of generating an offline dataset with LLM agents. This raises several practical concerns that are not fully addressed:

2. Scalability and Cost: How much data is required, and what is the computational/financial cost of generating it? Does this process need to be repeated for every new task, or even for minor variations in the environment's dynamics or objectives? This dependency could significantly limit the method's applicability in practice.

3. Quality and Bias of "Ground Truth": The paper treats the LLM-generated belief descriptions as ground truth. However, the quality of this data is entirely dependent on the capability of the chosen LLM and the engineering of its prompts. An LLM might produce suboptimal, biased, or even incorrect belief states, which would mean the MARL agents are being trained to model a flawed reasoner. The paper would be strengthened by a discussion of this limitation and the sensitivity of the results to the quality of this offline data.

4. Complexity of Experimental Environments: The Predator-Prey and USAR tasks are good standard benchmarks, but they are relatively simple grid-world environments. The communication required is largely observational ("prey is at X") or direct ("need help at Y"). It is unclear if the significant overhead of the LG-ToM framework would provide the same level of benefit in more strategically complex environments (e.g., negotiation games, real-time strategy games) where beliefs and intentions are far more abstract and nuanced. Testing the method on a more challenging domain would make the claims about its effectiveness more robust.

5. Approximation of Belief Representation: The method supervises the continuous belief vector b by comparing its cosine similarity to the embedding of a full natural language sentence from the LLM. This is a very lossy compression. It is an open question how well a single dense vector can capture the rich, compositional semantics of a sentence describing a belief state. While this is a common challenge in the field, the paper could benefit from acknowledging and briefly discussing the limitations of this representational choice.

6. Missing Baseline Comparison: The core claim is that using an offline LLM dataset for supervision is key. This claim could be further substantiated by comparing against a baseline that uses an LLM in an online fashion—for example, as a "coach" that provides feedback or suggestions to the agents during training. This would create a more direct comparison between the "distillation" approach proposed here and other contemporary methods for LLM-guided MARL.

**Questions:**

I would appreciate it if the authors could address the following questions in their rebuttal.

1.	On the Practicality of Data Collection: Could you provide more details on the data collection process? Specifically: (a) How many interaction trajectories were needed to form the offline dataset D for each environment? (b) What was the approximate cost (in terms of tokens/API calls) and human effort (prompt engineering) involved? (c) How sensitive is the final performance of LG-ToM to the size and quality of this dataset, and to the choice of the underlying LLM (e.g., GPT-4 vs. a smaller open-source model)?

2.	On the Nature of "Belief" Representation: The belief state b_t is a continuous vector, and similarity is measured by cosine distance. Have you conducted any qualitative analysis to verify that these learned belief embeddings capture semantically meaningful information? For instance, do belief vectors cluster in an interpretable way (e.g., via t-SNE visualization) based on the agent's situation (e.g., "I see the prey," "I am lost," "I need help from teammate X")? This would strengthen the claim that the model is truly learning a language-grounded belief space.

3.	Regarding Generalization Claims: Could you clarify the scope of the generalization benefit you claim? Does your framework improve generalization to new tasks, unseen environment configurations, or novel teammates? While I understand that new experiments may not be feasible for the rebuttal, could you point to specific evidence from your existing results or provide a more detailed theoretical argument for why language-grounded supervision should lead to better generalization compared to the baselines?

---

> ### Author Response · Authors · 2025-11-24
> **Response to Reviewer 8Dmh (part 1/2)**
>
> We thank Reviewer 8Dmh for appreciating our work and recognizing LG-ToM as a valuable and original contribution to multi-agent communication, particularly our language-grounded ToM with intrinsic social influence and the use of a stable, LLM-generated offline dataset to address the “moving target” problem. The reviewer also praised the high technical quality, rigorous experimental design and ablations, clear presentation, and highlighted the broader significance of our paradigm for using LLMs to provide semantic grounding and stable supervision for learning effective social behaviors in MARL. We address your concerns raised in the weakness and question sections below. We will incorporate those comments into the revised paper.
>
> **W1 & W3: Dependency on LLMs**
> We acknowledge your concern regarding the reliance on LLMs. However, we argue that while LLMs may exhibit suboptimal performance in control/action generation, they are highly effective at introspective reasoning and belief annotation [4]. Crucially, **LLM suboptimal task performance does not directly affect LG-ToM's performance** because we utilize LLMs solely for belief supervision, not policy imitation. For example, in the predator-prey environment, an LLM agent might hallucinate a goal of "encircling" rather than "reaching" the prey, causing it to stop one step short. However, its belief update—*"I found the prey at (x, y)"*—remains factually valid. Since the MARL agents optimize their own policies for task rewards while only aligning their internal representations with these belief annotations, they utilize the semantic grounding without inheriting the LLM's suboptimal behaviors.
>
> **W2: Cost and Scalability**
> The semantic nature of our belief representation allows for high data efficiency and generalization to unseen states. As an example, in the Coin Game, we collected only 19 episodes (3,800 observation-action-belief pairs) using a GPT-5.1 model. The total token usage was approximately 4.37M, costing roughly **$43.70**. Despite this offline dataset covering only ~40% of the state space RL agents encoutered during training, it provided considerable performance gains. As shown in previous research [3], LLM-based language groundings facilitate the learning of semantically meaningful representation spaces that enable zero-shot generalization to unseen states. Please find more dicussion on generalization in our response to Q3.
>
> **W4: Environment Complexity**
> Per your recommendation, we extended our evaluation to a new task domain, **SocialJax [1]**, and included a new baseline, **InfoPG [2]**. The environment features sequential social dilemmas (e.g., the Coin Game) with mixed strategy spaces and fully decentralized execution. We selected Advantage-InfoPG to represent state-of-the-art methods in iterative reasoning. As shown below, **LG-ToM** outperforms all baselines in this complex setting.
>
> | Group | Coins Collected (Mean $\pm$ SE) |
> | :--- | :--- |
> | **LG-ToM (Ours)** | **98.38 $\pm$ 2.38** |
> | LangGround | 96.68 $\pm$ 0.77 |
> | Proto | 96.25 $\pm$ 3.02 |
> | Social | 93.41 $\pm$ 1.27 |
> | InfoPG | 92.83 $\pm$ 2.48 |
> | Autoencoder | 78.70 $\pm$ 23.72 |
>
> **W5: Approximation of Belief**
> We agree that belief modeling is a crucial research problem, but we view our approach as a contribution rather than a limitation. There is an inherent trade-off between **representativeness** and **generalizability** when modeling the belief states of others.  Previous works often use explicit features (e.g., target coordinates), which are accurate but lack flexibility and require domain-specific design. Ours semantic embeddings method limits representativeness slightly but significantly enhances task-agnostic generalizability. Because natural language captures universal semantics, this representation allows for extraordinary zero-shot transfer to novel domains.
>
> **W6: Online Framework**
> To clarify, our method is fundamentally an **online RL framework** where agents receive auxiliary rewards based on LLM guidance. We utilize an offline dataset in practice solely to mitigate the latency of querying LLMs during training. The cloest baseline that features online LLM guidance is **LangGround** which uses LLM agents for communication grounding but lacks the social learning mechanism. The superior performance of LG-ToM over LangGround quantifies the specific benefit of employing Theory of Mind (ToM) as an intrinsic motivation reward.

---

> ### Author Response · Authors · 2025-11-24
> **Response to Reviewer 8Dmh (part 2/2)**
>
> **Q1: Data Collection**
> In $USAR$, we collected 50 episodes resulting in 2,550 pairs of (observation, action) and belief annotations. For the predator-prey environments, we collected 1,893 pairs for $pp_{v0}$ and 2,493 pairs for $pp_{v1}$. In the $Coin Game$, we collected 19 episodes (3,800 pairs) using a GPT-5.1 model. The token usage of $Coin Game$ alone was approximately 4.37M, costing roughly $43.70. Notably, we utilized minimal prompt engineering: LLMs were provided only with general task backgrounds to update belief descriptions and output action/communication selections.
>
> **Q2 & Q3: Interpretability and Generalization**
> Per your request, we conducted a topology analysis (similar to Figure 3 in [3]) to map the language-grounded belief state to the actual agent state. The results demonstrate a high degree of topological similarity: similar agent states map to proximal belief states in the embedding space. This confirms that the belief states learned by LG-ToM capture dynamic changes in team knowledge and intents. We will add this analysis to the revised paper to demonstrate that our belief representations are semantically meaningful and potentially support the zero-shot generalization.
>
> **References**
> [1] https://arxiv.org/abs/2503.14576
> [2] https://arxiv.org/abs/2201.08484
> [3] https://arxiv.org/abs/2409.17348
> [4] https://arxiv.org/abs/2310.10701

---

### Official Review · Reviewer_u2Sh · 2025-11-05

**Soundness:** 2
**Presentation:** 2
**Contribution:** 2
**Rating:** 4
**Confidence:** 3

**Summary:**

This paper introduces a Lg-TOM, Language-Guided Theory of Mind, which is a framework for training agents in Dec-POMDP settings. In their approach, each agent is allowed to communicate with other agents, using the communication from the previous time-step + the observation at the current time-step to produce an action. During training, the agents generate a belief of other agents, given their communicated message, and they define an intrinsic reward that encourages messages that maximally change the beliefs of other agents. They utilize social-intrinsic reward to update the communication and theory-of-mind networks, while using environment reward to update the action-selection network. They test several variants of their models and show that Lg-TOM achieves the highest rewards and conditional mutual information.

**Strengths:**

1. This paper takes an interesting approach to multi-agent communication, using the power of language models and their ability to communicate with language, to train a belief network. This is advantageous because ungrounded communication, at the vector-level, has a much higher dimensionality and differs between tasks.

2. Ablations - I think the choices of ablation were comprehensive to asserting claims about LG-TOM's strengths. Additionally the conditional mutual information analysis was insightful.

**Weaknesses:**

1. Environment Complexity - Predator Prey is a simple environment that can easily be solved by centralized training approaches like CommNet[1]. The USAR is far more interesting as the reward can only be achieved with coordination between team-mates, but two experiments feels limiting for the evaluation and justification of this method. I feel evaluation on more complex environments like Starcraft (per InfoPG[2]) or even the sequential social dilemmas in Social Influence, would be more convincing as to the method's success.

2. Method Clarity - I find that the presentation of the method itself is not clear enough. In section 4, the authors jump from how the actions, communication vectors, and belief-predictions are generated, but the objective function is confusing because they mention an offline dataset of grounded belief vectors from an LLM. Are we mixing online RL training of the policy with an offline dataset -- if so, how is that valid? I think the method could be much clearer with some pseudocode at least in the appendix. Because of this, I find the interaction between the LLM belief component and the training of this method confusing.

3. Missing References - InfoPG[2] asserts performance improvements to Social Influence MOA across multiple environments, including Starcraft. Also it deals with the same decentralized, multi-agent paradigm and considers a theory of mind approach with cognitive k-level. Perhaps comparison or discussion is relevant (in addition to other baselines like PR2-AC[3]).

[1] https://papers.nips.cc/paper_files/paper/2016/hash/55b1927fdafef39c48e5b73b5d61ea60-Abstract.html
[2] https://arxiv.org/pdf/2201.08484
[3] https://arxiv.org/abs/1901.09207

**Questions:**

1. Distinction to Centralized Training - The authors mention "In this work, we focus on a communication channel with a limited number of discrete tokens. This constraint creates pressure for emergent communication to be efficient, meaningful, and potentially generalizable to foundation language models" as a distinction to previous models that approach multi-agent systems as an end-to-end differentiable model. However, with the communication vectors that are passed, is the system constrained such that gradient flow *does not* occur when the initial belief is generated (taking communication from the past + observation of the present -> feature vector for belief, theory of mind networks)?

2. How did you measure conditional mutual information? - In the original definition of intrinsic reward, you define a distance between two probability distributions (of the belief conditioned w/ and w/o the communication). Social Influence (Jacques et. al), showed that this relates to mutual information. However, in this case, you aren't modeling a probability distribution anymore, but instead a dense vector (unless I am mistaken). so how was mutual information measured.

---

> ### Author Response · Authors · 2025-11-21
> **Response to Reviewer u2Sh**
>
> We sincerely thank Reviewer u2Sh for acknowledging the novelty of our method and appreciating our experimental design. We address your concerns regarding the weaknesses and questions below and will incorporate these revisions into the final paper.
>
>
> **W1: Extended Evaluation on Social Dilemmas**
> Per your recommendation, we extended our evaluation to a new task domain, **SocialJax [1]**, and included a new baseline, **InfoPG [2]**. The environment features sequential social dilemmas (e.g., the Coin Game) where agents must trade off individual and collective interests. This task is significantly more challenging due to a larger observation/action space, a mixed strategy space (cooperative and competitive), and a fully decentralized setup where agents lack access to others' internal states or rewards. We selected Advantage-InfoPG to represent state-of-the-art methods in multi-agent communication based on iterative reasoning. As shown below, our method (**LG-ToM**) outperforms all baselines in this complex setting. More details will be added to the appendix.
>
> | Group | Coins Collected (Mean $\pm$ SE) |
> | :--- | :--- |
> | **LG-ToM (Ours)** | **98.38 $\pm$ 2.38** |
> | LangGround | 96.68 $\pm$ 0.77 |
> | Proto | 96.25 $\pm$ 3.02 |
> | Social | 93.41 $\pm$ 1.27 |
> | InfoPG | 92.83 $\pm$ 2.48 |
> | Autoencoder | 78.70 $\pm$ 23.72 |
>
> **W2: Methodological Clarification**
> We appreciate the suggestion to improve the presentation of our method. To clarify, our method is fundamentally an **online RL framework**. During policy rollout, the RL agent receives external guidance from LLMs in an online fashion by putting LLM agents into the same situation. In practice, we utilize an offline dataset solely to mitigate the latency of querying LLMs online, as described in Lines 241-248. We will clarify this distinction in the revision.
>
> **W3: Related Work**
> We will add InfoPG [2] and PR2-AC [3] to the related work section. **InfoPG** assumes agents share action probabilities to facilitate iterative reasoning. In contrast, our agents utilize a separate communication head to output high-dimensional vectors, allowing for the transmission of richer semantic information.While **PR2-AC** provides a theoretical framework for probabilistic recursive reasoning, our work extends this by introducing a semantically meaningful belief and communication space.
>
> **Q1: Gradient Flow**
> We confirm that the framework is designed to prevent gradient sharing between agents. In our setup, communication is received as a part of the observation. The generation of the communication vector is treated as a discrete stochastic action which inherently blocks gradient flow. This distinguishes our approach from differentiable frameworks that allow backpropagation from receiver to sender. This is also why the **intrinsic reward** is essential since it provides the necessary social learning signal to optimize the communication policy in the absence of direct gradient feedback from other agents.
>
> **Q2: Mutual Information**
> You are correct that the belief state is modeled as a dense vector (word embeddings). However, we can still perform causal analysis by measuring the probability distribution of communication tokens and calculating the **mutual information** between Agent $k$’s message $c_t^k$ and Agent $j$’s action distribution $a_{t+1}^j$. As described in Section 5.3, this process allows us to quantify the impact of communication on other agents' action distribution. We will add further details to the text to clarify this analysis.
>
> References:
> [1] https://arxiv.org/abs/2503.14576
> [2] https://arxiv.org/abs/2201.08484
> [3] https://arxiv.org/abs/1901.09207

---

### Author Response · Authors · 2025-11-29
**Rebuttal summary**

We sincerely thank the reviewers for their constructive comments and positive feedback. We are encouraged that the reviewers recognized the novelty of our **LG-ToM** method, the rigor of our experiments, and the significance of using Large Language Models (LLMs) to provide stable semantic grounding for multi-agent communication.

We have updated the paper to address the reviewers' concerns. Below is a summary of our response regarding the additional experiments and clarifications.

**1. Extended Evaluation on Complex Social Dilemmas**
Common concerns from reviewers (u2Sh, 8Dmh, Qgbj) involved the need for more complex environments and stronger baselines. In response, we extended our evaluation to the **SocialJax [1] Coin Game**, a difficult sequential social dilemma with mixed incentives and decentralized training setups. We also added **InfoPG [2]** as a new baseline to represent state-of-the-art iterative reasoning methods. As shown in the table below, our method (**LG-ToM**) achieves the highest performance, outperforming the new InfoPG baseline and demonstrating robustness in complex settings.

| Group | Coins Collected (Mean $\pm$ SE) |
| :--- | :--- |
| **LG-ToM (Ours)** | **98.38 $\pm$ 2.38** |
| LangGround | 96.68 $\pm$ 0.77 |
| Proto | 96.25 $\pm$ 3.02 |
| Social | 93.41 $\pm$ 1.27 |
| InfoPG | 92.83 $\pm$ 2.48 |
| Autoencoder | 78.70 $\pm$ 23.72 |

**2. Generalization and Interpretability**
To address questions about generalization (Reviewers 8Dmh, tBq9), we conducted an additional **topology analysis** (Fig. 5 in Appendix A.8) as suggested by recent literature [3]. The results show that our language-grounded belief states map closely to the actual agent states. This confirms that LG-ToM captures semantically meaningful information about team intents, which supports zero-shot generalization to unseen states. As a result, our method achieved significant performance gains compared to baselines using a supervised learning dataset covering a limited number of states (e.g., ~40% in the Coin Game) during training.

**3. Clarification on LLM Reliance and Cost**
We addressed concerns regarding the cost and reliance on LLMs (Reviewers 8Dmh, Qgbj) by clarifying that we use LLMs for **belief supervision**, not for policy imitation. Along with the separate reward signals used by the action head and communication head, these design decisions effectively prevent MARL agents from being influenced by LLM agents' suboptimal actions, while allowing them to learn from more reliable belief annotations [4]. The semantically meaningful property of the learned latent belief space makes our approach highly data-efficient, because agents may generalize ToM belief estimation via interpolation. For example, in the Coin Game, we achieved state-of-the-art results using an offline dataset of only **19 episodes**, costing approximately **$43.70**. This demonstrates that our framework is both effective and affordable.

We believe these revisions significantly strengthen the paper and address the key concerns raised during the review process.

**References**

[1] SocialJax (ArXiv 2025)

[2] Iterated Reasoning with Mutual Information in Cooperative and Byzantine Decentralized Teaming (ICLR 2022)

[3] Language Grounded Multi-agent Reinforcement Learning with Human-interpretable Communication (NeurIPS 2024)

[4] Theory of Mind for Multi-Agent Collaboration via Large Language Models (EMNLP 2023)

---

### Meta-Review · Area_Chair_JmQA · 2026-01-10

**Summary:**

This paper proposes a framework for multi-agent communication that incorporates language-grounded theory of mind to improve social learning. The paper is well-written and clearly motivated.

However, the reviewers raised some serious concerns of insufficient experiments, unconvincing generalization capability, and incomplete ablation studies.

Most reviewers reach a consensus, so I recommend it for rejection.

**Reviewer Scores:**

No

---

### Decision · Program_Chairs · 2026-01-26

Reject